# Endemic Goiter and Iodine Prophylaxis in Calabria, a Region of Southern Italy: Past and Present

**DOI:** 10.3390/nu11102428

**Published:** 2019-10-11

**Authors:** Cinzia Giordano, Ines Barone, Stefania Marsico, Rosalinda Bruno, Daniela Bonofiglio, Stefania Catalano, Sebastiano Andò

**Affiliations:** 1Department of Pharmacy, Health and Nutritional Sciences, University of Calabria, 87036 Rende (CS), Italy; cinzia.giordano@unical.it (C.G.); ines.barone@unical.it (I.B.); stefania.marsico@unical.it (S.M.); rosalinda.bruno@unical.it (R.B.); 2Centro Sanitario, University of Calabria, 87036 Rende (CS), Italy

**Keywords:** iodine deficiency, iodine prophylaxis, goiter, urinary iodine concentration

## Abstract

Iodine, a micronutrient that plays a pivotal role in thyroid hormone synthesis, is essential for proper health at all life stages. Indeed, an insufficient iodine intake may determine a thyroid dysfunction also with goiter, or it may be associated to clinical features such as stunted growth and mental retardation, referred as iodine deficiency disorders (IDDs). Iodine deficiency still remains an important public health problem in many countries, including Italy. The effective strategy for the prevention and control of IDDs is universal salt iodization, which was implemented in Italy in 2005 as a nationwide program adopted after the approval of an Italian law. Despite an improvement in the iodine intake, many regions in Italy are still characterized by mild iodine deficiency. In this review, we provide an overview of the historical evolution of the iodine status in the Calabria region, located in the South of Italy, during the past three decades. In particular, we have retraced an itinerary from the first epidemiological surveys at the end of the 1980s to the establishment of the Regional Observatory of Endemic Goiter and Iodine Prophylaxis, which represents an efficient model for the surveillance of IDDs and monitoring the efficacy of iodine prophylaxis.

## 1. Introduction

Iodine deficiency and related disorders are still a public health problem that affects most countries, including industrialized and developing regions [1,2]. At the end of 2018, a global survey on iodine status, covering more than 97% of the world’s population, indicated that 21 countries remain vulnerable to iodine deficiency. Specifically, nationally-representative surveys revealed insufficient iodine intake in 14 countries (Burkina Faso, Burundi, Finland, Haiti, Israel, Iraq, the Democratic People’s Republic of Korea, Lebanon, Mali, Madagascar, Mozambique, Samoa, Vanuatu, and Vietnam). Moreover, in sub-national surveys, seven other countries, including Angola, Italy, Morocco, Norway, Russia, South Sudan, and Sudan were reported as being iodine insufficient [3].

Iodine deficiency impairs thyroid hormone production and has many adverse effects during the course of life, which are collectively termed the iodine deficiency disorders (IDDs). The frequency and severity of IDD manifestations are related to the degree of iodine deficiency and the age of the affected subjects. Although thyroid enlargement (goiter) is the classic sign of iodine deficiency, and can take place at any age, the most serious adverse effects of iodine deficiency occur during pregnancy, including impaired fetal growth and brain development [4,5,6]. The iodine status of the population can be assessed by using four methods: urinary iodine (UI) concentration, the goiter rate, serum thyroid stimulating hormone (TSH), and serum thyroglobulin (Tg) levels [7,8,9,10]. UI is the most sensitive indicator of current iodine intake because >90% of dietary iodine is excreted in the urine [11]. UI concentration can be measured in spot urine samples and the median UI values were used to assess iodine nutrition among school-age children, as recommended by World Health Organization (WHO), United Nations International Children’s Emergency Fund (UNICEF), and the Iodine Global Network (IGN) [7]. The cut-off values for urinary iodine levels were used to define iodine deficiency (<100 μg/L) were classified as mild (50–99 µg/L), moderate (20–49 µg/L), or severe (<20 µg/L). Daily iodine intake for population can be extrapolated from UI concentration using the following formula: urinary iodine (μg/L) × 0.0235 × body weight (Kg) = daily intake (μg) [12]. 

This allows us to assume that a median UI concentration of 100 μg/L corresponds roughly to an average daily iodine intake of 150 μg.

An important indicator of IDDs is represented by the goiter rate measured by ultrasound in school-age children. However, a standardized approach should be adopted worldwide to improve the reliability of thyroid volume in the context of IDD monitoring [8]. Supplementary indicators of iodine deficiency include blood-spot TSH measurement only in neonates [13], while Tg measured in a dried blood spot has been reported to be a good marker of iodine intake in infancy [14].

Universal salt iodization is the most cost-effective strategy for IDDs and the WHO, UNICEF, and IGN recommend that iodine is added at a concentration of 20–40 mg per kg salt, dependent on local salt intake [15].

Over the last decades, intensive efforts have been made by the governments of IDD-affected countries to implement and control salt iodization program [9,16,17,18,19]. India was one of the first countries in the world to initiate and maintain a sustained increase in the coverage of adequately iodized salt, achieving the goal of universal salt iodization levels of greater than 90% in urban areas of the Central, North, and North-East zones of its territory in 2015 [20,21,22]. Following the introduction of mandatory salt iodization in 1995, Madagascar showed a swift growth in iodized salt coverage, however a recent national survey reported that iodine deficiency remains a serious public health problem there [23]. This implies that to maintain an effective program on salt iodization over the long term, it is necessary to set up a system that coordinates and monitors the sale trend of iodized salt and communicates the health benefits of consuming iodized salt. A good example of national progress is represented by Ethiopia in which the national coverage of iodized salt increased from 4.2% in 2005 to 95% in 2014. These results stem from multi-level and multi-sector efforts involving public-private partnerships that focused on enforcing iodization legislation [24,25]. Also in Italy, a nationwide salt iodization program was implemented in 2005 with the approval of the law n. 55/2005 that requires the addition of potassium iodate to table salt at 30 mg/kg and the mandatory availability of iodized salt in food shops and supermarkets. The law also permits the use of iodized salt in the food and catering industries. To the aim of evaluating the efficiency and effectiveness of the nationwide program of iodine prophylaxis, in 2009 the Italian National Observatory for Monitoring Iodine Prophylaxis (OSNAMI) was established at the Italian National Institute of Health [26]. Although a significant improvement of iodine nutrition has been observed over the years, some regions in Italy still remain at risk of deficiency. 

In this review we provide an overview of the iodine status in Calabria, a region of Southern Italy, over the past three decades. Particularly, we report data obtained from the first epidemiological surveys up to the establishment of the Regional Observatory of Endemic Goiter and Iodine Prophylaxis (Figure 1), that represents an efficient model for the surveillance of IDDs and monitoring the efficacy of iodine prophylaxis.

## 2. History of Goiter and Iodine Deficiency in Calabria: Epidemiological Surveys during the 1980–2000 Period

The Calabria region located in the Southern of Italy is a peninsula of irregular shape, referred to as the “toe” of the Italian “boot”, with a coastline of 738 km on the Ionian and Tyrrhenian coasts of the Mediterranean Basin. The regional orography highlights mountainous features: 42% of the land is mountainous, 49% is hilly, and only 8% is completely flat with an average elevation of 597 m [27]. This region, comprising five provinces with a total population of about 2 million inhabitants, has been historically exposed to iodine deficiency. 

Data obtained from epidemiological surveys carried out between 1987–1996 allowed us to draw a first map of iodine deficiency and endemic goiter in Calabria by evaluating the mean of UI excretion and the size of thyroid gland. Since ultrasonography was not easy to perform in a large scale of epidemiological screening and the reference values for ultrasound thyroid volume measurement in children living in iodine-sufficient areas were not well established, the goiter rate was assessed using WHO’s 1960 palpation system [28].

The first study was conducted by our research group [29] in 1987–1989 on 34 villages of extraurban areas of Catanzaro (A) and Cosenza (B) provinces. In this survey, 4468 and 2721 schoolchildren (aged between 6–12 years) of area A and area B, respectively, were examined. The prevalence of endemic goiter was 53% in the population living in Catanzaro’s province with the highest percentage found in Zagarise (67%), while the rate in schoolchildren from Cosenza’s province was 44% with the highest percentage found in Laino Castello (69%). Interestingly, in both areas the goiter prevalence was independent from area altitude as well as the distance of the villages from the main town, and was significantly higher than that observed among the 1170 age-matched schoolchildren living in the urban area of the Calabria region (7.7%). Mean UI concentration was 49.7 ± 5.3 μg/L and 70.7 ± 3.1 μg/L in area A and B, respectively, indicating the presence of a mild iodine deficiency respect to the UI values of urban area that reflect an iodine sufficiency (104 ± 6.6 μg/L).

After two years of voluntary iodine prophylaxis (1991–1992) 855 schoolchildren from five small villages (Laino Borgo, Laino Castello, San Basile, Saracena and Mormanno) of Cosenza province were examined. These five villages were chosen since their drug-stores carried iodized salt and the population was advised to use it. As shown in Figure 2, an increase of UI concentration along with a reduced goiter prevalence were found, suggesting that an effective program of iodine prophylaxis is urgently needed in this region [29].

Another epidemiological study, which are useful to better define the map of endemic goiter and to characterize iodine deficiency in the whole Calabrian territory, was performed during the years 1990–1996 by Costante et al. [30]. A total of 13,984 schoolchildren, aged 6–14 years, was examined for the goiter prevalence, while UI excretion was evaluated in 284 samples that were randomly collected. Goiter prevalence ranged from 19% to 64% and from 5.3% to 25.7% in the inland territory and at the coastal area, respectively, while the mean of UI excretion was 53.8 ± 43.4 μg/L in the inland territory and 89.6 ± 59.8 μg/L at sea level, confirming that moderate levels along with pockets of severe iodine deficiency was present in the inland region, while iodine supply varied from sufficient to marginally low in the coastal areas. Moreover, the presence of mild to moderate iodine deficiency was also established by the results of neonatal TSH levels from the congenital hypothyroidism regional screening program. Indeed, the authors reported 14.8% frequency of TSH levels >5 μU/mL in newborns from the inland territory and 14.1% frequency from coastal areas [30].

Overall, these first epidemiological surveys clearly indicate that at the end of the 1990s, the whole Calabria region was a mild to moderate iodine deficient area. Interestingly, the data obtained in the Calabria region were in line with a series of surveys carried out from 1978 to 1991 in different regions of Italy, in which a total number of 72,112 schoolchildren was examined, including 5046 controls living in urban areas and 66,066 subjects residing in rural endemic areas. Surveys were carried out throughout Italy in predominantly hilly and mountainous areas. Globally, the goiter prevalence ranged from 14% to 73%, which inversely correlated with urinary iodine excretion (10–122 μg/g creatinine) and was more prevalent in Central and Southern Italy (reference [31] and references therein). 

At the end of the 1990s, as part of a European project entitled “Eradication of endemic goiter and of disorders of iodine deficiency in Southern Italy” with cooperation between the National Research Council and the Ministry of Health, and financed by the European Union, a survey to assess the iodine nutrition was conducted in eight regions of Southern Italy, including Calabria [32]. The grade of iodine deficiency was assessed through the measurement of UI excretion in 23,103 samples randomly collected from the schoolchildren population aged 11–14 years living in urban and rural areas and in different geographic locations. Median UI excretion in the all studied population was 74 μg/L, showing significantly higher values in urban areas compared to rural areas (81 μg/L vs. 73 μg/L, *p* < 0.0001). Besides, median UI excretion was significantly lower in inland mountainous/hilly areas respect to coastal mountainous/hilly areas (68 μg/L vs. 79 μg/L, *p* < 0.0001). The results of this extensive survey indicated that in Southern Italy, mild to moderate iodine deficiency still persisted [32]. Particularly in the Calabria region, data obtained from a total of 2693 spot urinary samples expressed as median as well as mean (±SD) displayed insufficient iodine intake in all five provinces of the Calabria region (Table 1, data unpublished).

Similar results were obtained in Campania, another region of Southern Italy, in which UI excretion from 10,552 schoolchildren were determined. The analysis of frequency distribution showed values below 50 and 100 μg/L of UI in 32% and 61% of children, respectively, highlighting the Campania region as a mild iodine deficiency area [33].

As a part of the same European project, another important challenge was to implement the use of iodized salt through interactive meetings with schoolchildren. In the Calabria region, we have interviewed 49,840 subjects in their classrooms, providing detailed information on the beneficial effects of iodine salt prophylaxis along with informative materials consisting of leaflets and table-games about iodine deficiency disorders. A final goal of this project was to establish an Observatory for Monitoring Iodine Prophylaxis in each Italian region.

## 3. Status of Iodine Intake Over the Last Two Decades in the Calabria Region: The Epidemiological Observatory for Endemic Goiter and Iodine Prophylaxis

Based on our extensive studies carried out in the entire regional territory and taking into account the final goal of the European project, the Epidemiological Observatory and Promotion of Health of the Calabria Region, Section “Goiter Endemic and Iodine Prophylaxis” (OERC) was established by the Calabria region (regional law n. 755/2003) at the Health Center of the University of Calabria. The OERC represents the epidemiological structure through which the regional-scale surveillance of the iodine prophylaxis program is carried out using: (i) epidemiological surveys to periodically evaluate the iodine intake and the prevalence of goiter in the adolescent and to verify the prevalence of thyroid diseases in the adult population; (ii) a promotional campaign on the advantages of iodine prophylaxis; (iii) the sale trend of iodized salt.

### 3.1. Epidemiological Surveys

The first survey was carried out in the years 2007–2009 on 11–14 year old children recruited from long standing iodine sufficient urban areas (U) and from rural areas (R) in which an iodine insufficiency was previously documented [29,30]. In agreement with the guidelines of WHO, UNICEF, and International Council for Control of Iodine Deficiency Disorders (ICCIDD) [9], monitoring was based on both the percentage of goiter and the median value of UI concentration in schoolchildren. A total of 2733 subjects (1686 U and 1047 R) from the five provinces of Calabria were examined to evaluate thyroid volume by ultrasonography, while 1358 (794 U and 565 R) spot-urine samples were collected to determine adequacy of iodine intake. The prevalence of goiter, calculated on the basis of the reference values proposed by WHO [8], and the median values of ioduria are shown in Figure 3 (data unpublished).

Our data indicated that a mild iodine deficiency was still present in the rural areas of the provinces of Cosenza, Reggio Calabria, and Crotone, whereas all the areas in which the prevalence of goiter was less than 10% showed an adequate iodine status. 

On the basis of these findings, we focused our attention on a vast territory of mild to-moderate endemic area of Cosenza province, including four villages (Laino, San Basile, Saracena, and Mormanno). In particular, we assessed both goiter prevalence and UI concentration in children aged 11–14 years. Using WHO criteria, the goiter prevalence was 7.1% and 10.95% normalized for body surface area (BSA) and age, respectively, while median UI excretion was 113 μg/L. Moreover, we also evaluated the efficacy of the iodine prophylaxis in an adult population living in Laino. We observed reduced goiter prevalence in the studied population that was subjected for two decades to a program of salt iodization. More interestingly, the beneficial effects of iodine prophylaxis were also observed in the youngest adult population investigated (ranged 18–27 years), which showed almost an absence of thyroid enlargement, whereas the older adult population (>58 years), which have mostly lived in a severe iodine deficient area before beginning iodine supplementation, were less responsive in reducing goiter prevalence [34].

The epidemiological studies on iodine status by OERC continued until 2012 in the context of the activities carried out by OSNAMI. Median UI concentration and goiter prevalence in 729 schoolchildren recruited in Calabria region were 87 μg/L and 7.5%, respectively, which showed that further efforts were still required to encourage the use of iodized salt [35]. 

Despite the clear benefits of iodine prophylaxis, continuous surveillance of adverse effects induced by iodine intake needs to be carefully maintained. Thus, the frequency of thyroid disorders along with the levels of antithyroid antibodies (TgAb and TPOAb) in 560 adult subjects from Laino and from the urban area of Cosenza were evaluated [36]. As expected, the prevalence of subjects affected by goiter was significantly higher in the rural area than in the urban area, but interestingly it was significantly lower compared with that reported in the adult population living in the same rural area in 2007 (42.6% rural area in 2007 vs. 13.8% rural area in 2015, *p*< 0.0001). Moreover, we have observed a significant increase of TgAb levels in subjects living in a long-standing iodine sufficient area that may be an epiphenomenon with no pathogenic significance [37]. Interestingly, no changes were detected for concentrations of TPOAbs, the levels of which are typically high in thyroid autoimmune disease [38]. 

More recently, preliminary data from the national program of iodine deficiency monitoring activities of OSNAMI have reported an increased UI concentration in rural areas as well as urban areas together with a reduction in goiter prevalence in schoolchildren population of most Italian regions, including Calabria [39].

### 3.2. Promotional Campaign

An intense and widespread iodine prophylaxis campaign was carried out during 2007–2009. At the end of the promotional survey, the OERC medical team visited 1012 primary schools involving more than 100,000 children in all provinces of Calabria, outlining the health benefits of iodine through the distribution of information materials, pamphlets, gadgets, and posters. Besides, other strategies to increase consumer awareness towards iodized salt and its beneficial health effects have been developed and are currently ongoing. These include a promotional campaign using mass media (newspaper, TV), billboards on buses, and a website [40] that offers a useful platform containing national and international links to other reliable sources of information about iodine nutrition.

### 3.3. Sale Trend of Iodized Salt

The activities of the OERC also include an assessment of the iodine content in salt on the market. Salt samples, taken from the subjects screened (131 subjects from the rural area and 235 subjects from the urban area) during our epidemiological surveys, were found to be compliant with the iodine content permitted by Italian law (30 mg/kg) [36]. The data on sale trends of iodized salt in Calabria, supplied by the Italian Salt Company, one of the most important sales producers/distributors in the region, showed an increasing trend over the last few decades, reaching a coverage rate of approximately 65% [36]. These data are in line with those obtained from the national salt producers and collected by the Italian National Institute of Health that specifically reported an increase in the percentage of sold iodized salt from 34% in 2006 to 65% in 2017 [39]. However, since a usage rate of iodized salt of at least 90% is recommended by WHO, UNICEF, and the ICCIDD [3] for effective prevention of IDDs, further efforts should be made to better inform the population on the benefits of using iodized salt. 

## 4. Conclusions

The IDD control program in Calabria is one of the success stories of public health in Italy. The epidemiological data over the last three decades clearly indicate the improvement of iodine status in the Calabria region also due to the commitment of the Regional Observatory of Endemic Goiter and Iodine Prophylaxis. Although substantial progress has been made, efforts should focus on ensuring there is adequate iodine intake in the entire population to achieve and maintain the IDD control goal.

## Figures and Tables

**Figure 1 nutrients-11-02428-f001:**
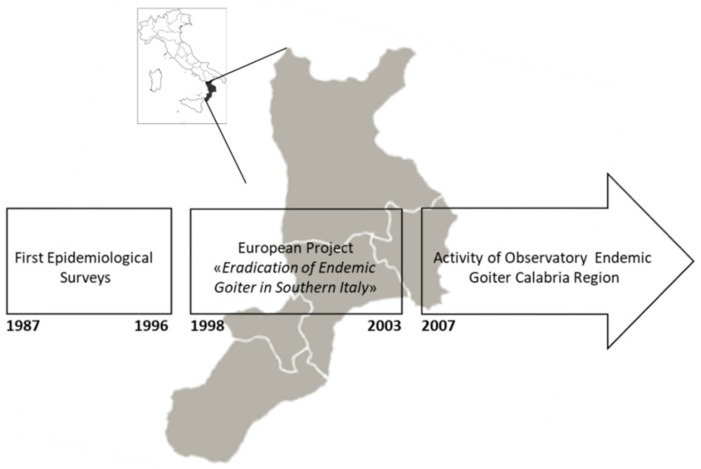
Schematic representation of the key steps of epidemiological surveys conducted from the late 1980s to the present for assessing and monitoring iodine status in the Calabria region.

**Figure 2 nutrients-11-02428-f002:**
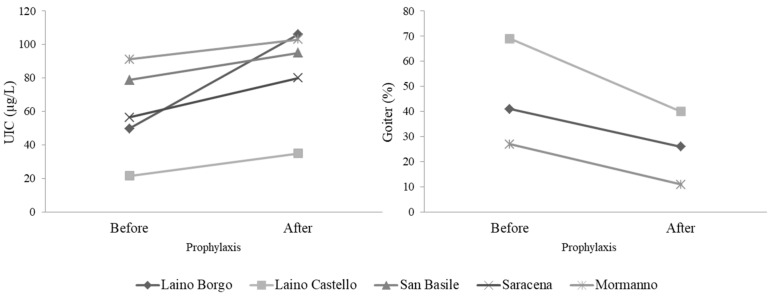
Urinary iodine concentration (UIC) and goiter prevalence before and after iodine prophylaxis in schoolchildren of the Cosenza province.

**Figure 3 nutrients-11-02428-f003:**
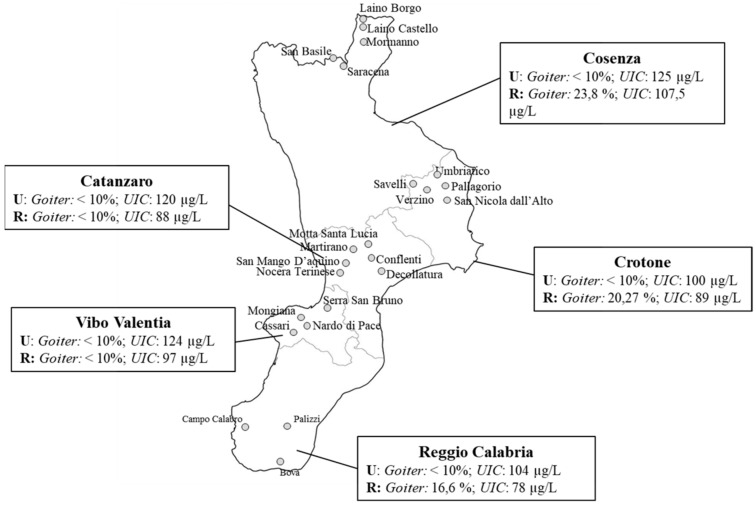
Goiter prevalence and median urinary iodine concentration in schoolchildren population from iodine sufficient urban areas (U) and rural areas (R) of the Calabria region.

**Table 1 nutrients-11-02428-t001:** Mean (±DS) and median urinary iodine concentration (UIC) in schoolchildren in the Calabria region.

Provinces	Samples (n)	UIC Mean (±SD) μg/L	UIC Median μg/L
Catanzaro	1024	85 ± 71	65
Cosenza	701	91 ± 71	73
Crotone	257	84 ± 78	54
Reggio Calabria	346	91 ± 69	75
Vibo Valentia	365	83 ± 64	67

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
