# Peer review of "Endemic Goiter and Iodine Prophylaxis in Calabria, a Region of Southern Italy: Past and Present"

_nutrients, 2019, doi:10.3390/nu11102428_

Round 1

Reviewer 1 Report

Insufficient supply of iodine in the human diet is a global problem that affects about 2 billion people. The study discusses the important topic of preventive iodine supplementation in areas affected by its deficiency. As in many countries, in Italy the main source of this element is iodized salt. The positive impact of this process on reducing the incidence of thyroid diseases have already been proven many times in numerous countries around the world.
The topic is interesting, while the text presented by the authors does not meet the requirements of scientific work. It is also not a review of the literature because the authors present some data obtained in their research. In addition, the number of the cited literature certainly does not exhaust the topic discussed in the study. Rather, it is data that could be compiled in local or part of national statistical yearbooks.
In my opinion, recognizing the presented text as a scientific publication and accepting it for publication in journal would require significant changes in the experimental part, which is small in its current form. The authors limited themselves to providing several data obtained in extensive studies. However, they do not present in the text or research methodologies (even how to analyze the obtained samples), they do not subject the results to any statistical analysis. The authors also did not include in the text any discussion or comparison of the impact of salt iodization in Italy with the experience of other countries. In addition, most of the presented results come from research obtained in 2007-2009 or earlier, which cannot be considered acctualy at the current rate of science development.
When authors discuss the results of commercially available salt samples, the only statement is their compliance with applicable Italian law. The authors do not present basic data here (e.g. the number of samples tested, how they were collected, the method of analysis, not to mention the presentation of detailed results or statistical analyzes, etc.).

               For the above reasons, it seems difficult to adapt the presented research results to the requirements of a scientific publication or literature review which would be necessary to accept this publication.

Reviewer 2 Report

This manuscript is good an example of the countries that have iodine deficiency problem and it needs to be followed up for the health safety of their population.   If the authors can add inform of following up program to control iodine deficiency in this area after 2009 by government or any organization.  It will be perfect for this manuscript.

The authors also can improve the quality of Figure 1, 2 and 3 because it really hard to read information from these figures.

Thank you for the good manuscript on micro-nutrient program

Reviewer 3 Report

On page 2 line 46 it should be specified that thyroglobulin and TSH measurement in neonates are actually useful for monitoring iodine deficiency.

A revision of the english text is recommended.

Round 2

Reviewer 1 Report

If the journal accepts 'Communication', which is not a literature review or own research, I have no reservations.